# Prevalence and Etiological Characteristics of Norovirus Infection in China: A Systematic Review and Meta-Analysis

**DOI:** 10.3390/v15061336

**Published:** 2023-06-07

**Authors:** Ting-Ting Li, Qiang Xu, Mei-Chen Liu, Tao Wang, Tian-Le Che, Ai-Ying Teng, Chen-Long Lv, Guo-Lin Wang, Feng Hong, Wei Liu, Li-Qun Fang

**Affiliations:** 1School of Public Health, Guizhou Medical University, Guiyang 550025, China; ltt202307@163.com; 2State Key Laboratory of Pathogen and Biosecurity, Beijing Institute of Microbiology and Epidemiology, Beijing 100071, China; xq_bime@163.com (Q.X.); kdrery@163.com (M.-C.L.); wang_tao0515@126.com (T.W.); wanzhuanmeitian@outlook.com (T.-L.C.); tay0001@163.com (A.-Y.T.); tijys2012@163.com (C.-L.L.); guolin_wang2019@163.com (G.-L.W.); 3School of Public Health, Anhui Medical University, Hefei 230032, China

**Keywords:** norovirus, etiological characteristics, meta-analysis, China

## Abstract

Norovirus is a common cause of sporadic cases and outbreaks of gastroenteritis worldwide, although its prevalence and the dominant genotypes responsible for gastroenteritis outbreaks remain obscure. A systematic review was conducted on norovirus infection in China between January 2009 and March 2021. A meta-analysis and beta-binomial regression model were used to explore the epidemiological and clinical characteristics of norovirus infection and the potential factors contributing to the attack rate of the norovirus outbreaks, respectively. A total of 1132 articles with 155,865 confirmed cases were included, with a pooled positive test rate of 11.54% among 991,786 patients with acute diarrhea and a pooled attack rate of 6.73% in 500 norovirus outbreaks. GII.4 was the predominant genotype in both the etiological surveillance and outbreaks, followed by GII.3 in the etiological surveillance, and GII.17 in the outbreaks, with the proportion of recombinant genotypes increasing in recent years. A higher attack rate in the norovirus outbreaks was associated with age group (older adults), settings (nurseries, primary schools, etc.) and region (North China). The nation-wide pooled positive rate in the etiological surveillance of norovirus is lower than elsewhere in the global population, while the dominant genotypes are similar in both the etiological surveillance and the outbreak investigations. This study contributes to the understanding of norovirus infection with different genotypes in China. The prevention and control of norovirus outbreaks during the cold season should be intensified, with special attention paid to and enhanced surveillance performed in nurseries, schools and nursing homes from November to March.

## 1. Introduction

Norovirus is a common cause of sporadic gastroenteritis in all age groups and is a major cause of acute gastroenteritis outbreaks in humans, accounting for approximately 50% of acute gastroenteritis outbreaks worldwide [1]. Human infection with norovirus brings a significant global disease burden, with about 685 million infections worldwide, and more than 200,000 deaths [2], USD 4.2 billion in direct health system costs, and USD 60.3 billion in social costs every year [3]. Norovirus infection is also an important cause of diarrhea death in children under 5 years of age, suggesting a significant threat to children [4]. Noroviruses have shown genetic diversity in 10 genomes (GI–GX) according to polymerase and capsid gene sequences, of which the GI, GII, GIV, GVII and GIX genomes can infect humans [5]. About 90% of global norovirus outbreaks and sporadic cases are associated with the GII genogroup [6]. Seasonality concerning outbreaks has been reported, with the majority occurring in cool months [7]. Common symptoms of norovirus infection include vomiting, diarrhea, abdominal pain, fever, chills, headache, and myalgia [8]. Norovirus is a self-limiting disease that usually causes mild symptoms [9], but it can lead to serious consequences in children, older adults, and other immunocompromised people infected with norovirus [10,11]. 

In China, norovirus has also caused a severe disease burden with about 20% of acute gastroenteritis cases attributed to its infection, and GII.4 and GII.3 have been reported as the dominant genotypes [12]. During 2014–2017, norovirus infection became the most common cause of reported outbreaks of acute gastroenteritis, resulting in more than 30,000 cases reported by the National Public Health Emergency Event Surveillance System (PHEESS) [13]. Nurseries and schools are major settings for norovirus outbreaks and pose a high risk to children and adolescents [13,14]. Most norovirus outbreaks are caused by human-to-human transmission, and some recent outbreaks have been associated with newly emerging recombinant genotypes [14]. Recent studies have described the situation of norovirus outbreaks as epidemic based on epidemic surveillance systems, such as the Public Health Emergency Event Surveillance System and CaliciNet China [13,14]. Fan Yu et al. conducted a systematic review on the epidemiological characteristics, transmission mode and genotype distribution of norovirus outbreaks in China from 2000–2018 [15], suggesting that the epidemiological and clinical characteristics vary between regions, seasons, and genotypes. However, there is a lack of research on genotype distribution and the clinical characteristics of norovirus infection in etiological surveillance in combination with outbreak investigation, and no studies have focused on the factors that influence the outbreak attack rate on a national scale.

In order to strengthen the understanding of the epidemiology and clinical characteristics of human norovirus infection with different genotypes, and to guide the surveillance of sporadic cases and the control of outbreaks caused by norovirus, we conducted a systematic review of the reported outbreaks and etiological surveillance of norovirus infection in China. This study aims to reveal the epidemiological and clinical characteristics and dynamics of norovirus infection with different genotypes, as well as the patterns of norovirus outbreaks in different populations, seasons, regions and settings, and to analyze the factors that influence the attack rate of norovirus outbreaks.

## 2. Materials and Methods

This study was registered with the International Prospective Register of Ongoing Systematic Reviews (PROSPERO), CRD42022297065, and was conducted according to the Preferred Reporting Items for Systematic Reviews and Meta-Analyses (PRISMA) statement (Appendix A) [16].

### 2.1. Search Strategy and Selection Criteria

A literature search was performed on four databases using a set of terms and Boolean operators, including the PubMed database (https://pubmed.ncbi.nlm.nih.gov/, accessed on 1 April 2021), China National Knowledge Infrastructure (CNKI, http://www.cnki.net/, accessed on 1 April 2021), Wanfang Database (http://www.wanfangdata.com.cn/, accessed on 1 April 2021), and Chongqing VIP Chinese Science and Technology Journal Database (CQVIP, http://www.cqvip.com.cn/, accessed on 1 April 2021). Different retrieval formulas were developed for these databases based on individual retrieval methods. The standardized medical subject heading (MeSH) term “Norovirus”, “Norwalk Virus”, and free word “China” were used for the PubMed database, while “Norovirus”, “Norwalk virus” and “Norwalk-like virus” were set as the subject heading, title, and keywords to search in CNKI, Wanfang, and CQVIP (Appendix A). All articles published between January 2009 and March 2021 were searched without language restrictions.

Comprehensive inclusion and exclusion criteria were pre-defined to facilitate the screening process. To be included, published reports possessed the following characteristics: (1) the study subjects lived in the mainland of China; (2) noroviruses were reported as the etiologic agent; (3) the study period was between 2009 and 2021; (4) the types of articles included etiological surveillance and outbreak investigation. Reports of the laboratory methods, animal and plant studies, vaccines, model studies, cross-sectional studies in healthy people, health education, molecular mechanisms, and reviews related to norovirus were excluded from the final analysis (Appendix A). The deduplication and addition of the articles was carried out using Endnote X9 software and manual methods. Titles and abstracts of the retrieved studies were screened using Endnote X9 independently by two reviewers (TTL and MCL) to identify studies that might be eligible for inclusion, and then the full texts of the potentially eligible studies were retrieved and independently assessed for eligibility by the two reviewers. Discrepancies between reviews were resolved by consensus or a third reviewer (QX). Studies potentially describing overlapping data were noted and the duplications were removed (e.g., same hospital and population during an overlapping time period).

### 2.2. Data Extraction and Variable Definitions

The full texts of all the included literature were reviewed and the data were extracted using a standardized form. A total of 20 variables were included (details in Appendix A). Based on the average monthly temperature of each city included in the study, the top six months were classified as the “warm season” and the rest as the “cold season” [17]. A norovirus outbreak was defined as >5 acute gastroenteritis cases within 3 days after exposure in a common setting where >2 samples (whole fecal, rectal swab, or vomitus) tested positive for norovirus. Acute gastroenteritis was defined as >3 events involving loose feces, vomiting, or both, within 24 h [14]. For articles containing information on more than one outbreak, the data of each outbreak were extracted separately. Moreover, outbreaks reported in multiple publications were recorded only once. The Microsoft Excel program (version 2019) was used for data entry.

### 2.3. Data Analysis

The descriptive statistics included a frequency analysis for the categorical variables, medians, and inter-quartile range (IQR) for the continuous variables. We further performed a meta-analysis to estimate the attack rate of the outbreaks, positive detection rate of norovirus for etiological surveillance, and the proportion of clinical manifestations for patients infected with norovirus. Briefly, the proportion of clinical manifestations was estimated by dividing the number of cases with a single clinical presentation by the total number of cases. Heterogeneity was estimated using the statistic of I², of which >50% was considered as significant heterogeneity. If substantial heterogeneity existed, a random effect model was used. Otherwise, the fixed-effects model was preferred to summarize the pooled percentage, as well as a 95% CI [18] (Appendix A). A subgroup analysis was performed in the meta-analysis to compare the categorical variables between groups. For the estimation of the attack rates in the subgroups, outbreaks with fewer than 30 at-risk individuals were excluded. The principles of the Transparent Reporting of Systematic Reviews and Meta-Analysis were implemented throughout the study. In addition, this investigation expanded upon previous research by exploring the association between norovirus infection and potential risk factors using the beta-binomial (BB) regression model; a *p* value <0.05 was considered statistically significant [19]. All data analyses were carried out using R version 4.0.3 (R Foundation for Statistical Computing, Vienna, Austria), meta-analyses were performed using R package “meta” and graphical presentations were performed using R package “ggplot2” and the ArcGIS 10.7 (Esri Inc., Redlands, CA, USA).

## 3. Results

### 3.1. Overview of the Publications and Spatiotemporal Characteristics of Patients

The literature search identified a total of 6198 articles. After discarding the duplicates, 3237 articles were screened by the title and abstract, of which 1713 articles were screened by the full text. A total of 1132 articles were included in this study according to the inclusion and exclusion criteria, with 632 articles on etiological surveillance and 500 articles on outbreak investigation (Figure 1, Appendix A).

A total of 155,865 confirmed cases were reported, with the number of cases increasing from 2009 and peaking in 2018 followed by a pattern of decline. The numbers of published articles and outbreak events also showed a similar pattern, both peaking in 2018 (Figure 2A). A pooled test positive rate of 11.54% (104,896 patients infected with norovirus) was shown among 991,786 patients with acute diarrhea based on the etiological surveillance, with no difference between North and South China. The highest positive rate was shown in adolescents (14.74%) and adults (14.74%), followed by children (14.17%) and older adults (14.05%). The pooled attack rate of norovirus was 6.73% based on the data of 899 outbreak events, with a higher attack rate in North than in South China (10.10% vs. 4.81%, *p* < 0.05). The largest number of outbreak events was found among adolescents, followed by children, adults and older adults, while the highest attack rate was found among older adults (11.85%), followed by children (9.48%), adolescents (5.53%) and adults (4.55%). Most of the norovirus outbreaks occurred in primary schools, followed by nurseries, secondary schools and universities. Human-to-human transmission was the main transmission model in the o utbreaks (44.65%), which occurred more frequently in the cold season than the warm season (Table 1, Appendix A). Most outbreak events were published during the period from 2017–2020, with 83.11% (246/296) in North China and 61.27% (367/599) in South China. The highest attack rate was shown in 2016 in North China (15.77%) and in 2012 in South China (7.69%) (Figure 3A, Appendix A). Based on the 866 outbreak events with monthly information, significant seasonality of norovirus outbreaks was shown, with a peak from November to March of the next year in South China (70.14%, 404/576) and dual peaks, respectively, from March to June and November to December in North China (80%, 232/290) (Figure 3B, Appendix A). Diarrhea and vomiting were the most common symptoms of norovirus infection in adults and other age groups, respectively, and these also varied between genotypes (details in Appendix A).

### 3.2. Demographic Characteristics of Patients in Norovirus Outbreaks

Most outbreaks occurred in adolescents (352 outbreaks), followed by children (158 outbreaks), adults (124 outbreaks) and older adults (13 outbreaks). Based on the 641 outbreak events with age and monthly information, the number of outbreaks peaked in October–November for children (70.14%, 53/157), November–May of the next year for adolescents (83.29%, 289/347), November–April of the next year for adults (75.81%, 94/124), and January for older adults (41.67%, 5/12). The highest attack rate was shown in May for children (17.70%) and adolescents (9.50%), in April for adults (10.83%), and in March for older adults (16.05%) (Figure 3C, Appendix A).

Based on the 870 outbreak events with setting and monthly information, the outbreaks mainly occurred in October–May of the next year in nurseries (89.11%, 180/202), November–May of the next year in primary schools (85.10%, 217/255), November–April of the next year in secondary schools (60.11%, 110/138), March and November–December in universities (50.82%, 31/61), February and April in restaurants (44.44%, 16/36), February in communities (31.25%, 10/32), January in workplaces (25%, 8/32), December–January of the next year in hospitals (48.15%, 13/27), November–January of the next year in nursing homes (75%, 12/16), and November–April of the next year in other settings (76.12%, 51/67). The highest attack rate was shown in January for workplaces (13.30%), February for nursing homes (26.50%), March for secondary schools (6.67%), April for other settings (13.47%), May for primary schools (11.10%), June for nurseries (17.03%) and universities (7.48%), July for hospitals (17.84%), September for restaurants (26.07%), and December for communities (17.14%) (Figure 3D, Appendix A).

### 3.3. Temporal and Geographic Pattern of Norovirus Genotypes in China 

Genotype information was reported in 24.91% (282/1132) of the articles (Appendix A), including 153 articles on etiological surveillance (13,343 cases) and 129 articles on outbreak investigations (183 outbreaks, 10,107 cases). Based on the etiological surveillance, the most common genotype was GII.4 (45.18%, 6028/13,343), followed by recombinant genotypes (22%, 2936/13,343) and GII.3 (11.92%, 1591/13,343). GII.4 was the predominant genotype from 2011 to 2018 (no data in 2009 and 2010). The proportion of recombinant norovirus increased significantly from 2013 to 2019, which was mainly attributed to GII.4[Pe], while GII.3 remained comparable between the years (Figure 2B). The most common genotype causing norovirus outbreaks was GII.17 (20.33%, 37/183), which was mainly reported in studies published from 2015 to 2017, followed by GII.4 (19.78%, 36/182), which was mainly reported in studies published from 2014–2016 and 2020, and recombinant genotypes (13.73%, 25/182), which were mainly reported in studies published from 2018–2020 (Figure 2B). The genotypes responsible for the largest number of cases in the outbreak events were GII.4 (2454 cases), GII.17 (2385 cases), and GII.12 (1543 cases), respectively. The GII.2[P16] was the dominant genotype attributing to the outbreaks caused by the recombinant genotypes (Appendix A). 

Based on the 182 outbreak events with geographic and genotype information, GII.4 was the predominant genotype causing the outbreaks in South China, 23.43% (30/128), while it was less common in North China. GII.17 was the predominant genotype causing outbreaks in both North China (18.52%, 10/54) and South China (21.09%, 27/128). The attack rates of the outbreaks caused by GII.12 were the highest in both North China (43.52%) and South China (17.85%) (Figure 4A, Appendix A). GII.17 caused the most cases (35.27%, 1284/3641) in 54 outbreaks in North China and GII.4 caused the most cases (32.72%, 2374/6285) in 128 outbreaks (Figure 4B). According to the different ecological regions in China, GII.17 was the dominant genotype in the North China region and Central China region, and GII.4 was the dominant genotype in the South China region. There were few reports with norovirus genotype information in the Northeast region and Inner Mongolia–Xinjiang region, and no outbreaks of norovirus have been reported in the Qinghai–Tibet region and Southwest region (Appendix A). 

Based on the 111 outbreak events with age and genotype information, the largest number of norovirus outbreaks were caused by recombinant genotypes in adolescents (14 outbreaks) and by GII.17 in adults (1276 cases) (Figure 4C). According to the 183 outbreak events with setting and genotype information, the largest number of norovirus outbreaks were caused by GII.2 in nurseries (11 outbreaks) and by GII.4 in universities (1334 cases) (Figure 4D).

### 3.4. Influencing Factors of Attack Rate in Norovirus Outbreaks

A total of 216 articles involving 302 outbreak events were included in this analysis. The attack rates varied from 0.06% to 88.89% between the different outbreaks. The beta-binomial model showed that norovirus outbreaks had lower attack rates in children (RR: 0.13, 95% CI: 0.03, 0.63), adolescents (RR: 0.09, 95% CI: 0.02, 0.35), and adults (RR: 0.18, 95% CI: 0.05, 0.65), compared to older adults. Norovirus outbreaks had higher attack rates at nurseries (RR: 4.43, 95% CI: 1.57, 12.55), primary schools (RR: 4.01, 95% CI: 1.98, 8.12), secondary schools (RR: 3.15, 95% CI: 1.55, 6.40), restaurants (RR: 4.47, 95% CI: 2.28, 8.79), workplaces (RR: 2.91, 95% CI: 1.81, 4.67), and other settings (RR: 3.93, 95% CI: 2.05, 7.51) than universities. The attack rate was higher in North China (RR: 1.97, 95% CI: 1.61, 2.41) than in South China. There was no significant difference in the other factors, e.g., the seasons and routes of transmission (Appendix A).

## 4. Discussion

In recent years, China has made progress in controlling the diarrheal diseases but the disease burden of norovirus infection remains high [12]. Therefore, we conducted an exhaustive literature collection of the published articles, explored the epidemiological and clinical characteristics and dynamics of norovirus infection caused by different genotypes in China, as well as the patterns of norovirus outbreaks in different populations, seasons, regions and places.

Our research showed that the nation-wide pooled positive detection rate (11.54%) in the etiological surveillance of norovirus was similar to that of the active surveillance of patients with acute diarrhea managed by the Chinese Centers for Disease Control and Prevention (12.47%) [17], which was lower than the global overall detection rate (18%) and that in developing countries (19%) [20]. Our results show that older adults have the highest attack rates during outbreaks and reports of norovirus outbreaks among elderly people in communities and nursing homes have become more frequent in recent years [21,22]. Most current global initiatives to reduce the burden of diarrhea focus on children under five years of age, so the burden of disease in older people is a growing public health challenge that will have increasingly negative consequences if the issue is ignored. Therefore, due attention is required.

Norovirus infection is also known as winter vomiting disease [23]. Our study shows that norovirus outbreaks occurred mainly from November to March, with the fewest in July and August. Our findings were consistent with the data of the Public Health Emergency Event Surveillance System and CaliciNet China, which showed that most norovirus outbreaks occur between October and March, while the outbreaks were less common in July and August [13,14]. The outbreak peaks were different in the North and South, but the lowest outbreak occurred in July and August both in the North and South. The monthly difference in norovirus prevalence between northern and southern China may be associated with temperature and rainfall [24,25]. In China, cooler temperatures from November to March play an important role in the spread of the virus. In general, there are more outbreaks in the cold season than in the warm season, and the global seasonal pattern of norovirus indicates that its epidemic peak is in winter, suggesting that there is some correlation between temperature and norovirus epidemics. At the same time, studies have also suggested that there is a positive correlation between rainfall and seasonality [7]. Different environments and settings also play a decisive role in the spread of norovirus. Nurseries, primary schools, secondary schools and universities were the main outbreak settings, which is consistent with the data from the Public Health Emergency Event Surveillance System and CaliciNet China, which show that norovirus outbreaks occur predominantly in nurseries and schools [13,14,15]. The government requires school staff to check and screen children attending nurseries and schools for fever, vomiting and diarrhea every morning and to report any infectious disease immediately [26]. This may be why nurseries and schools in China have seen the most outbreaks of norovirus. The peak of the outbreak was the same in nurseries, primary schools and secondary schools, but the outbreak peak in universities was different from the other three major sites. In the major settings, the fewest outbreaks occurred in July and August. The attack rate at universities is lower than elsewhere, which is supported by the beta-binomial model. Universities had the lowest attack rate of norovirus outbreaks, probably because university students and staff are healthy adults with occasional close contact, in contrast to nurseries, health care settings, etc., where sick individuals are either treated or are very young or very old and are taken care of, thus, have more close contacts.

GII.4 and the recombinant genotypes were the dominant genotypes in both the etiological surveillance and outbreak investigations, consistent with the surveillance results in several regions of China and in the global population [20,27,28]. GII.3 was mainly observed in etiological surveillance, and most outbreaks events were caused by GII.17. In some studies, non-GII.4 strains had replaced GII.4 as the predominant causes of norovirus outbreaks since 2014, while in this study, the phenomenon started in 2015, with the difference likely due to publication delays [15,29]. In addition, due to the publication delay, GII.17 was the predominant genotype in the norovirus outbreaks in China since 2015, while other studies have shown that GII.17 has been the major cause of norovirus outbreaks in China since 2014 [29,30]. In addition, consistent with the results of the Japanese study, a new norovirus variant, GII.17 [P17], has been prevalent in Japan since December 2014 and became the dominant genotype of norovirus outbreaks in March 2015 [31]. GII.2 [P16] was the recombination genotype that caused the most outbreaks of norovirus, which was consistent with CaliciNet China’s result [14]. In nurseries, norovirus outbreaks were dominated by GII.2, while recombinant genotypes dominated in the norovirus outbreaks in primary schools. In addition, GII.17 and GII.4 were the dominant genotypes in secondary schools and universities, respectively.

The main symptoms of norovirus infection are vomiting and diarrhea, which usually lasts a relatively short time. The clinical presentation of norovirus infection varies by age group and genotype. Diarrhea was the most common symptom among adult patients and vomiting was the most common symptom among patients in other age groups. For infection with different genotypes, the most common symptom was diarrhea in GII.4-infected patients and vomiting in GII.17-infected patients, the same as in the Swedish study [32]. In addition, it has been shown that infection with the GII.4 strain leads to more severe consequences than infection with a non-GII.4 strain [23].

There are some limitations to this study. First, our study suffers from publication bias. Most of these studies have been reported from South China, which may affect the comparability of norovirus infection characteristics between North and South China. Second, since our data are from different authors, and most outbreaks do not share the same variables, this limits the number of studies analyzed. We also excluded articles that did not mention specific information for each outbreak in the summary outbreak analysis, resulting in a reduction in the number of outbreaks included in the analysis. Third, the dual nomenclature of ORF1 and VP1 sequences can be used to identify recombinant noroviruses [5], but most of the norovirus genotypes in the literature we have included are determined with partial VP1 sequences, which prevents us from fully studying the recombinant noroviruses. This study contributes to the attainment of an enhanced understanding of norovirus infection in China, demonstrates the need to consider the wider surveillance of norovirus, thereby helping to inform future research and surveillance efforts, and provides an overview of norovirus epidemiology for future vaccine policy decisions.

## Figures and Tables

**Figure 1 viruses-15-01336-f001:**
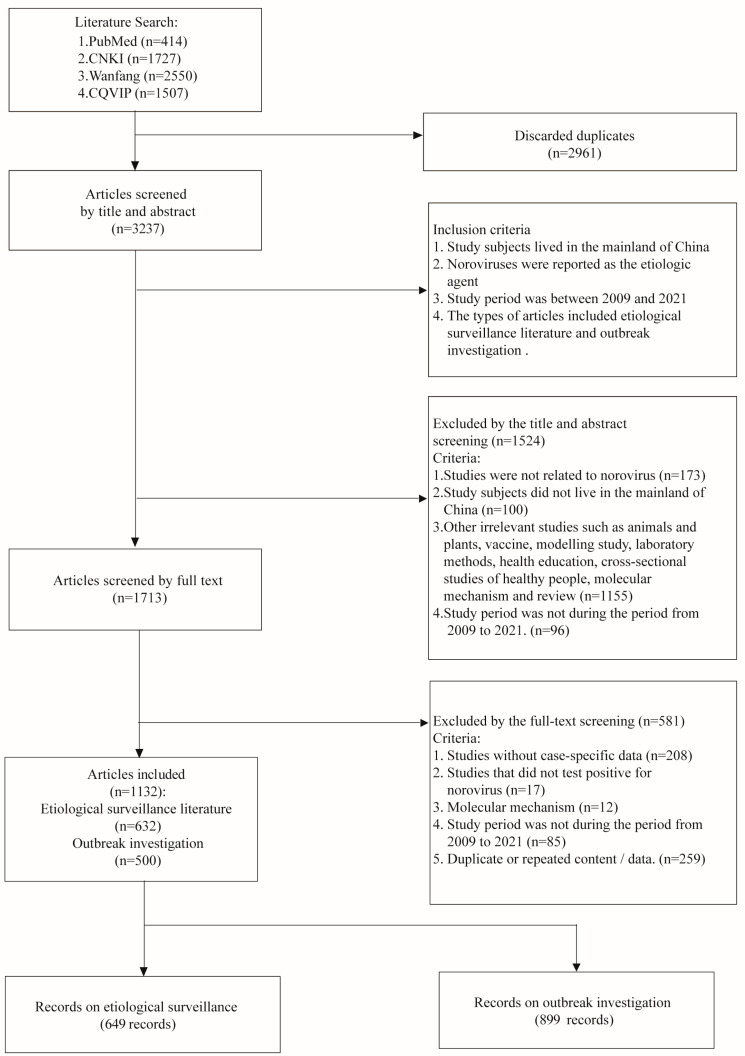
Flow diagram of the literature review. CNKI: China National Knowledge Infrastructure; Wanfang: Wanfang Database; CQVIP: Chongqing VIP Chinese Science and Technology Journal Database.

**Figure 2 viruses-15-01336-f002:**
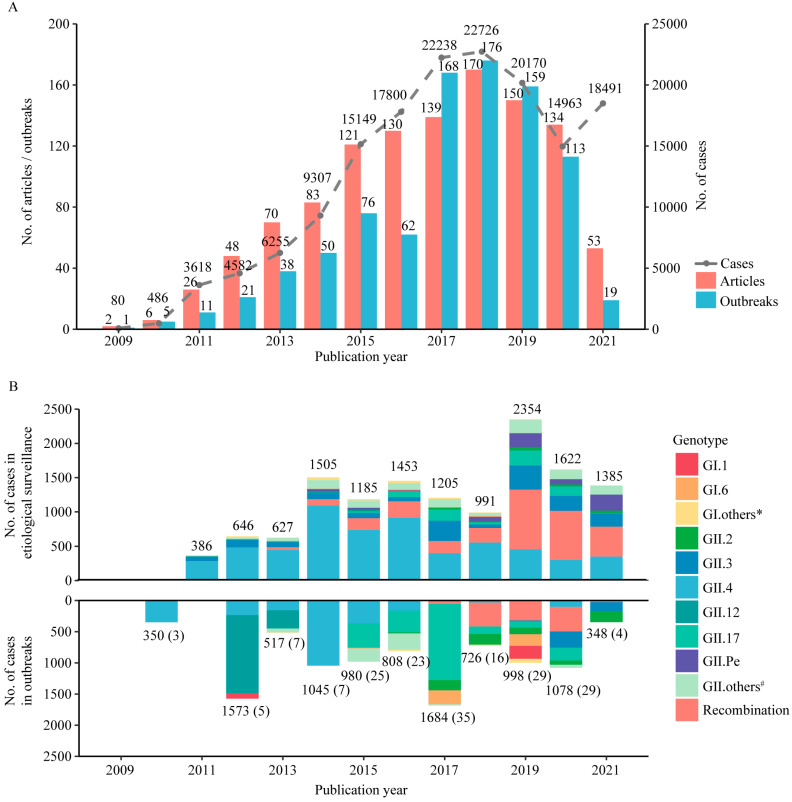
Temporal distribution of reported norovirus infections and genotypes in China. (**A**) number of articles, outbreaks and reported cases infected with norovirus over the publication year. (**B**) genotype of norovirus reported from studies on etiological surveillance and outbreak investigation over the publication year. Numbers in parentheses indicate the number of outbreaks. * Includes GI.2, GI.3, GI.4, GI.5, GI.7, GI.8, GI.9, and GI.Pd. # Includes GII.1, GII.P1, GII.P2, GII.P3, GII.P4, GII.5, GII.P5, GII.6, GII.P6, GII.7, GII.P7, GII.8, GII.P8, GII.10, GII.11, GII.P12, GII.13, GII.P13, GII.14, GII.15, GII.16, GII.P16, GII.P17, GII.21, GII.P21, GII.22, GII.P22, GII.P31, GII.b, GII.e, GII.g and GII.Pg.

**Figure 3 viruses-15-01336-f003:**
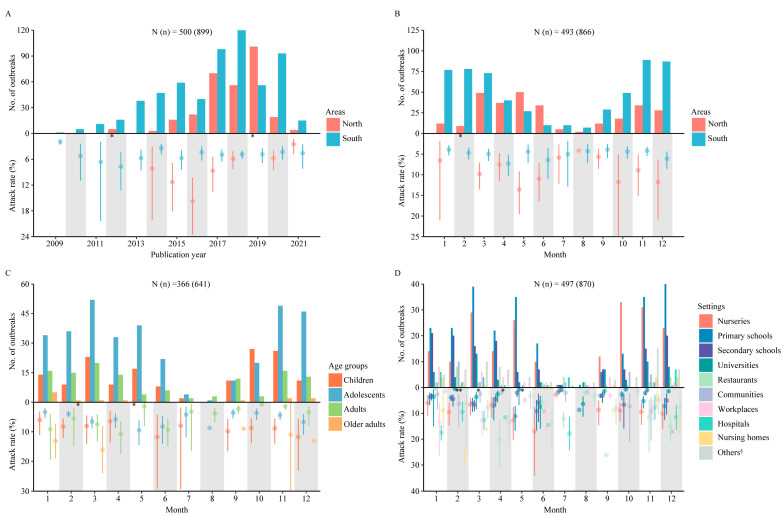
Characteristics of the distribution of norovirus outbreaks in China by region, age and setting. Number of outbreaks and attack rate estimated using meta-analysis based on outbreak investigations between North and South China by year (**A**), month (**B**), monthly number of outbreaks and attack rate estimated using meta-analysis based on outbreak investigations by age (**C**) and setting (**D**). N (*n*) indicates the number of articles (outbreaks) included in this analysis. For the attack rate, the mean and a 95% CI are presented. * Indicates either a quite high mean attack rate or a wide 95% CI, which was presented in North China in 2012 (3.79% [95% CI: 0.42, 26.85]), in 2019 (17.39% [95% CI: 14.46, 20.76]), in February (17.68% [95% CI: 9.34, 30.93]), in older adults in February (27.85% [95% CI: 17.96, 37.73]), in children in May (17.70% [95% CI: 9.44, 30.73]), at restaurants in February (12.58% [95% CI: 2.86, 41.30]) and March (12.90% [95% CI: 0.70, 75.79]), at a community in May (40.54%), at workplaces in February (79.41%) and April (20.01% [95% CI: 8.36, 40.71]). ^§^ Indicates households, transportations, welfares, military settings, prisons, multiple setting types and multiple school types on panel D.

**Figure 4 viruses-15-01336-f004:**
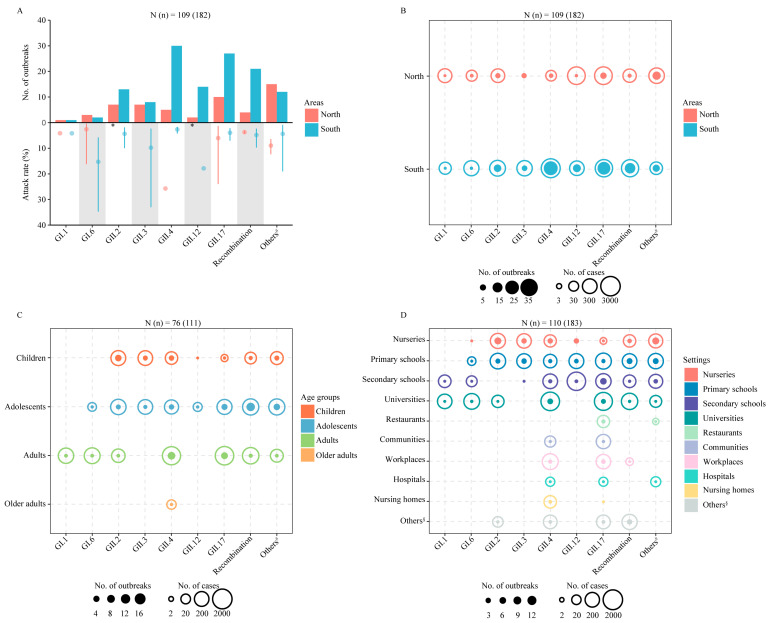
Genotype characteristics of norovirus outbreaks in China by region, age and setting. The monthly number of outbreaks and the attack rate were estimated using a meta-analysis based on the outbreak investigation by virus genotypes (**A**), and the numbers of outbreaks and reported cases for different norovirus genotypes norovirus by region (**B**), by age (**C**), and setting (**D**). N (*n*) indicates the number of articles (outbreaks) included in this analysis. For the attack rate, the mean and a 95% CI are presented. * Indicates either a quite high mean attack rate or a wide 95% CI, which was presented for GII.2 in North China (15.94% [95% CI: 4.37, 44.04]), for GII.12 in North China (43.52%). ^#^ Indicates GI.2, GI.3, GI.5, GII.6, GII.7, GII.8, GII.14, and GII.21 on panel A, panel B, and panel D and GI.3, GI.5, GII.6, GII.7, GII.8, GII.14, and GII.21 on panel C. ^§^ Indicates households, transportations, welfares, military settings, prisons, multiple setting types and multiple school types on panel D.

**Table 1 viruses-15-01336-t001:** Characteristics of human norovirus infection among etiological surveillance and outbreaks in China.

	Etiological Surveillance				Outbreak Investigation			
	No. of Articles	Cases (Median, IQR)	Detection Positive Rate (95% CI)	*p*	No. of Articles(No. of Outbreaks)	Cases (Median, IQR)	Attack Rate (95% CI)	*p*
**Total**	632	70.00 (34.00, 156.00)	11.54 (10.79, 12.33)		500 (899)	26.00 (14.00, 65.00)	6.73 (6.11, 7.40)	
**Age groups (year)**				0.996				<0.001 *
Children (0–4)	91	64.00 (26.00, 137.25)	14.17 (12.05, 16.61)		70 (158)	14.00 (9.00, 19.00)	9.48 (7.76, 11.53)	
Adolescents (5–17)	31	76.00 (26.50, 198.50)	14.74 (9.82, 21.52)		209 (352)	28.00 (15.00, 60.00)	5.53 (4.82, 6.34)	
Adults (18–60)	17	58.50 (26.50, 138.25)	14.74 (9.82, 21.53)		114 (124)	64.00 (25.00, 115.00)	4.55 (3.39, 6.08)	
Older adults (≥60)	5	13.00 (5.00, 20.00)	14.05 (9.40, 20.47)		12 (13)	11.00 (9.00, 21.00)	11.85 (7.58, 18.06)	
**Settings**								<0.001 *
Nurseries	–	–	–		102 (212)	15.00 (10.00, 20.00)	9.13 (7.70, 10.80)	
Primary schools	–	–	–		136 (265)	24.00 (13.00, 55.00)	6.07 (5.10, 7.20)	
Secondary schools	–	–	–		104 (140)	50.00 (25.75, 81.00)	4.19 (3.46, 5.06)	
Universities	–	–	–		56 (61)	97.00 (63.50, 197.50)	1.78 (1.24, 2.55)	
Restaurants	–	–	–		31 (36)	16.00 (8.00, 29.25)	9.46 (4.80, 17.81)	
Communities	–	–	–		32 (32)	50.00 (25.75, 100.75)	5.93 (3.72, 9.32)	
Workplaces	–	–	–		30 (32)	43.00 (21.00, 83.25)	9.62 (5.96, 15.16)	
Nursing homes	–	–	–		15 (16)	18.00 (10.00, 25.50)	8.99 (5.67, 13.95)	
Hospitals	–	–	–		25 (27)	17.00 (12.50, 22.50)	13.16 (10.84, 15.88)	
Others	–	–	–		60 (78)	52.50 (25.25, 101.00)	6.24 (4.53, 8.55)	
**Seasons**				<0.001 *				0.218
Cold	43	7.00 (3.75, 18.25)	20.92 (19.50, 22.35)		357 (593)	27.00 (14.00, 67.50)	5.85 (5.20, 6.58)	
Warm	43	5.00 (2.00, 12.00)	14.95 (13.26, 16.65)		164 (273)	25.00 (13.00, 63.00)	6.62 (5.65, 7.75)	
**Areas**				0.586				<0.001 *
North	228	61.50 (31.00, 133.25)	11.27 (10.06, 12.60)		116 (296)	19.00 (12.00, 40.00)	10.10 (8.58, 11.85)	
South	401	77.00 (36.00, 176.00)	11.72 (10.78, 12.73)		380 (599)	31.50 (15.00, 76.00)	4.81 (4.31, 5.36)	
**Routes of transmission**								0.032 *
Human-to-human	–	–	–		125 (213)	18.00 (13.00, 40.00)	6.33 (4.97, 8.04)	
Foodborne	–	–	–		108 (125)	57.50 (21.75, 95.25)	7.70 (6.51, 9.10)	
Waterborne	–	–	–		78 (84)	82.50 (45.00, 149.25)	5.21 (4.05, 6.66)	

“–” indicated no data. Etiological surveillance sites were all in hospitals and no transmission routes were involved; * Significant at *p* < 0.05; Subgroup analysis was performed in the meta-analysis to compare the categorical variables between the groups.

## Data Availability

Data will be made available upon request made to the corresponding author.

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
