# Peer review of "Prevalence and Etiological Characteristics of Norovirus Infection in China: A Systematic Review and Meta-Analysis"

_viruses, 2023, doi:10.3390/v15061336_

Round 1

Reviewer 1 Report

Quite an interesting general paper for the years to come

General comments:

Could you think how this information will change your thinking on strategies of preventing norovirus outbreaks/emergence in China? Enhancing surveillance in institutions? Certain timing/places? If any concrete suggestions, please add to the manuscript. A bit more specific than "special attention paid to high-risk age groups and settings"

Also, please compare the Chinese findings to those elsewhere globally in your discussion and add one summary sentence to abstract.

Minor

- l. 48 "winter" is different in various parts of the globe. Could you say cold season/cold months. Please think and revise as necessary, or define winter=November to March (or whatever),

l. 68-70. Sentence is missing something?

l. 309. Vaccination of what? Covid-19?

l. 326-328. "Universities had the lowest attack rate of norovirus outbreaks, probably because university students have strong herd immunity" Or maybe because university students and staff are healthy, adults with occasional close contact in contrast to nurseries, health care settings etc. where either sick individuals are treated or very young or very old are taken care of, thus with more intimate contacts.

l. 346 onwards. Which one was it? Was it the genotype that caused diarrhoea or vomiting, or the age group?

Reviewer 2 Report

Overall, the paper was well designed and presented.  It is more like a reviewer paper.  Author might want to specify the terms of pooled attack rate and detection rate.  They looked low.  The paper was based upon database and published papers.  It will be better to include the reported cases and outbreaks from Chinese CDC to see if they are consistent. From the Figure, it looks like the reported cases, outbreaks, and published papers reached the peak in 2018 and all figures dropped after 2018.  As they are from the published papers and database submission, all the dropped cases and outbreaks were associated with the dropped publications.  Therefore, I am wondering if the cases and outbreaks from Chinese CDC supported this.  
